# RE-PERG in early-onset Alzheimer's disease: A double-blind, electrophysiological pilot study

Alberto Mavilio[1]*, Dario Sisto[2], Florenza Prete[3], Viviana Guadalupi[3], Rosanna Dammacco[2], Giovanni Alessio[2]

1 Social Health District, Glaucoma Center, Azienda Sanitaria Locale–Brindisi, Brindisi, Italy, 2 Department of Neurosciences, Institute of Ophthalmology, University of Bari, Bari, Italy, 3 Social Health District, Alzheimer Evaluation Units, Azienda Sanitaria Locale—Brindisi, Brindisi, Italy

* a.mavilio@gmail.com

## Abstract

### Purpose

To evaluate the ability of re-test pattern electroretinogram (RE-PERG), a non-invasive and fast steady-state PERG, to detect inner retinal bioelectric function anomalies in patients with early-onset Alzheimer's disease (AD).

### Methods

The study population consisted of 17 patients with AD-related mild cognitive impairment (MCI), 16 patients with vascular dementia (VD)-related MCI, both assessed using the neuropsychological Mini-Mental State Examination (MMSE) and by structural magnetic resonance imaging, and 19 healthy, age-matched normal controls (NC). All participants were visually asymptomatic, had normal or near-normal general cognitive functioning and no or minimal impairments in daily life activities. Visual field (VF) test, optical coherence tomography (OCT) and RE-PERG, sampled in five consecutive blocks of 130 events, were performed.

### Results

There was no statistically significant difference among the three groups with respect to age, VF parameters (mean and pattern standard deviations) and OCT parameters (ganglion cell complex thickness and retinal nerve fiber layer thickness). The mean amplitude in the RE-PERG was significantly lower, but only weakly in the AD group than in NC (p = 0.1) whereas the intrinsic variability of the 2nd harmonic phase was significantly higher in the AD group than in either the VD or NC group (p<0.001).

### Conclusions

RE-PERG is altered in early-stage AD, showing a reduced amplitude with high intrinsic phase variability. It also allows the discrimination of AD from VD. A high intrinsic variability in the PERG signal, determined using RE-PERG, may thus be a new promising test for neurodegenerative diseases.

**Data Availability Statement:** All relevant data are within the manuscript.

**Funding:** The Authors received no specific funding for this work.

**Competing interests:** The Authors have declared that no competing interest exist.

## Introduction

Alzheimer's disease (AD), the most common type of dementia, is characterized by the extracellular accumulation of amyloid-β protein (Aβ) plaques and intraneuronal aggregates of hyperphosphorylated tau that form neurofibrillary tangles in the brain. AD develops in ~5% of individuals over 65 years of age and in about 20% of those over 85 years of age. Currently, AD affects 26 million people around the world, and by 2050 over 100 million are expected to be affected. [1] A rare, early-onset familial AD has also been reported. [2] A non-specific cognitive decline, referred to as mild cognitive impairment (MCI), may precede AD and is frequent in the elderly population. [3,4] In addition, there are several recognized risk factors for AD, including diabetes, obesity and hypercholesterolemia. [5] The diagnosis of AD is currently made using a series of tests, beginning with questionnaires, such as the Mini-Mental State Examination, designed to assess the intellectual, emotional and functional status of the patient. [6] Second-level tests include positron emission tomography (PET), single photon emission computed tomography (SPECT), cerebrospinal fluid Aβ42 level measurement, and assessment of medial temporal lobe atrophy via brain magnetic resonance imaging (MRI). [7]

Worsening of visual function is a common feature of AD, [8] and the accumulation of Aβ plaques and aggregates of hyperphosphorylated tau in the visual association cortices, [9,10] primary visual cortex, [11,12] lateral geniculate nuclei, [13,14] and the retina [15–17] has been reported. The visual disturbances in AD were long considered to be due to damage in the primary and associative visual cortex, but a primary involvement of retinal ganglion cells (RGCs) and their axons has also been proposed. [15,18–19]

Furthermore, AD patients may suffer deficits in contrast sensitivity. [20–22]

The visual pathway is composed of two different systems. The magnocellular (M) system recognizes achromatic stimuli. It originates from large RGCs and projects first to the magnocellular layers of the lateral geniculate nucleus and then to lamina 4C-α of the visual cortex. The second system is specific for color discrimination. It originates from small RGCs and projects first to the parvocellular layers of the lateral geniculate nucleus and then to lamina 4C-ß of the visual cortex. This system can be further divided in two different color-based pathways: the red-green parvocellular (P) and the blue-yellow koniocellular (K) subsystems. The M system responds to achromatic stimuli, and the P subsystem to chromatic stimuli but also to achromatic contrast stimuli of high spatial frequency. However, within the same range of spatial frequencies, M-cells are more sensitive to achromatic stimuli, especially at higher temporal frequencies. [23] Whether AD specifically affects one or the other sub/system is unclear. Pathologies in both the M system and the P subsystems have been described in the lateral geniculate nucleus and in the retina, [23–25] but other evidence suggests a specific M pathway involvement. [23,25–28]VD is the second most common type of dementia worldwide, and its prevalence in individuals age 65 and older is expected to double every 5 years. [29] It leads to several cognitive disorders as well as behavioral and locomotor abnormalities. The most important cause of VD is cerebral small-vessel disease; other causes are cardiac and carotid atherosclerosis, cardioembolism, hypertensive vasculopathy, aneurysm, vascular malformations, amyloid angiopathies, monogenic disorders involving stroke as well as metabolic, hematological and vasospastic disorders. Although, like AD, a diagnosis of VD can be made with certainty only post-mortem, strong clinical suspicion is based on history, timing of the event, cardiovascular and hematological assessment, psychometric evaluation and neuroimaging features. [29]

In the evaluation of AD, neuroimaging techniques include structural MRI and PET (tracing amyloid, fluorodeoxyglucose, tau). The typical MRI features of AD are a reduction of gray-matter volume, cortical atrophy and a reduced hippocampal volume. Amyloid PET is

recommended especially in patients with otherwise unexplained cognitive impairment or an atypical clinical presentation. Other types of PET are mainly used in clinical research. [30]

In VD, typical imaging features are white matter lesions, cortical and subcortical infarctions and intracerebral microhemorrhage. Extensive parenchymal infarctions are due to large-artery disease, and small infarctions especially to small-vessel disease. [29] In the eye, the primary involvement in AD patients is the RGCs [15–19] whereas the visual disturbances found in VD are often due to cerebral infarctions involving the optic pathway, leading to typical visual field alterations according to the affected site; retrogeniculate alterations do not determine subsequent optic atrophy [31] Primary involvement of the retina has also been documented in patients with cerebral autosomal arteriopathy with subcortical infarcts and leukoencephalopathy, [32] hereditary endotheliopathy with retinopathy, nephropathy and stroke, cerebroretinal vasculopathy and hereditary vascular retinopathy, which are interpreted as different phenotypes of the same disease, i.e., autosomal dominant retinal vasculopathy with cerebral leukodystrophy. [33–35] In all of these diseases, retinal damage is due to vascular retinopathy, not to primary neurodegeneration. Unlike other parts of the central nervous system (CNS), RGCs are relatively accessible and can be studied both anatomically and functionally to obtain information related to the state of neurons, including in patients with AD. [36] The properties of RGCs are similar in many ways to those of brain neurons such that anomalies in these cells can be related to brain dysfunction. In patients with AD both optic nerve degeneration and a loss of ganglion cells have been demonstrated. [37–39]

The pattern electroretinogram (PERG) is an electrophysiological test used to assess RGCs function. [40,41] Although developed for the early diagnosis of glaucoma, its utility in neurological diseases, including multiple sclerosis, [42] AD [23, 43–45] and Parkinson's disease, [46] all of which are characterized by inflammation, neurotransmission anomalies, and neurodegeneration, has also been demonstrated. PERG can provide useful diagnostic, prognostic and follow-up information on these diseases.

A specific form of PERG is steady-state PERG (SS-PERG), in which a fast (steady-state) stimulus generates a sinusoidal response that can be analyzed by Fourier transform. This allows the isolation of a second harmonic whose amplitude and phase delay can be evaluated. The PERG amplitude is related to the number of surviving neurons, and the PERG phase delay to synaptic dysfunctions of living neurons. [47] Synaptic damage and remodeling of the RGCs dendritic tree have also been histologically demonstrated in mouse models of glaucoma. [48,49] However, while a reduced amplitude is observed in patients with glaucoma and in those with ocular hypertension, [50–52] it is also a feature of conditions not related to glaucoma, such as cataract and myopia. [53–55] To overcome the limits of SS-PERG, a new test, the re-test PERG (RE-PERG), was recently introduced for the more accurate diagnosis of glaucoma. It is based on five consecutive SS-PERG stimulations without pause and evaluates the individual-intrinsic within-test phase variability of the second harmonic, rather than strictly the amplitude. Phase variability was shown to be very low in healthy controls but the standard deviation of the phase is higher in glaucoma patients. [56] Moreover, unlike the amplitude, phase variability is not influenced by optical media opacities and myopia. [57,58] Second-level imaging-based tests for the diagnosis of AD and VD are often expensive and not always available, especially in rural hospitals, such that diagnostic tools based on biomarkers able to distinguish among the various types of dementia are needed. RE-PERG uses high temporal frequency stimuli able to evoke a response of the M system. A higher phase variability is related to RGCs dysfunction, which precedes ganglion cells loss. Thus, the current research evaluated the ability of RE-PERG to detect anomalies in the primary inner retinal bioelectric function of M-cells in patients with early-onset AD compared to those with VD and in NC.

## Materials and methods

From September 1st to December 15th 2018, 52 consecutive patients (33 with MCI and 19 age-matched, healthy controls were finally enrolled in this study. All patients were recruited at the Alzheimer Evaluation Units of the Brindisi Social Health District, Brindisi, Italy. Neurologic exclusion criteria were: neurological/psychiatric conditions other than mild AD and VD, antidepressant-antipsychotic medication, history of malignancy, head trauma or stroke, drug abuse or addiction and metabolic or endocrine anomalies.

Ophthalmic exclusion criteria were: diabetes even in the absence of retinopathy, [59] ocular hypertension and glaucoma as diagnosed by the EGS guidelines, [60] congenital optic nerve head anomalies, retinopathy or any other ocular or general condition or therapy that might influence visual function, a best corrected visual acuity <20/40 (Snellen acuity), spherical refraction >±5.0 D, cylinder correction >±2.0 D and optic media opacities. The healthy control (HC) group consisted of age- and sex-matched healthy individuals with no evidence of dementia as reported by the participant or his/her family.

### Assessment of cognitive function

In the neuropsychological evaluation, cognitive function was assessed using MMSE, a simple screening test that measures global cognitive function [61] by assessing orientation, memory, concentration, language, and design capacity. The same experienced examiner administered the test. The MMSE total score ranges between 0 and 30, with lower scores indicating a poorer cognitive ability. [62] Scores ≥28 points indicate normal cognition and <28 points mild (24–27 points), moderate (10–23 points) or severe (≤9 points) cognitive impairment. A score of ≤9 points is considered to be almost diagnostic of dementia. [63]

All patients underwent structural MRI. AD and VD were diagnosed according to international consensus criteria. [64]

### Ophthalmic examination

Each participant underwent a comprehensive ophthalmic evaluation, including a review of medical history, best-corrected visual acuity testing, IOP measurement by Goldmann applanation tonometry, ultrasound pachymetry (Pachmate GH55 DGH Technology, Inc. Exton PA, USA), slit-lamp biomicroscopy, gonioscopy, and dilated fundus examination with a 78 lens. The criteria for the clinical and instrumental ophthalmic evaluation were the same as used in previous studies. [56–58]

### Standard Automated Perimetry (SAP)

The visual field was assessed using a Humphrey field analyzer, model 745i II (Carl Zeiss Meditec, Germany) and the 24–2 SITA standard strategy. Near addition was added to the refractive correction value. If fixation losses were >20% and false-positive or false negative results >15%, the test was repeated. At least two SAPs were performed to ensure reliable results and minimize the effect of learning. [65]

### Spectral-domain Optical Coherence Tomography (OCT)

Peripapillary retinal nerve fiber layer (RNFL) and ganglion cell complex (GCC) thicknesses were assessed using a Zeiss Cirrus HD OCT-500 (software version 7.0.1.290, Carl Zeiss Meditec, Dublin, CA). The protocol's 200 × 200 optic disc cube was used to perform a circular scan 3.46 mm in diameter. The scan was automatically targeted around the optic disc to provide the RNFL thickness of the four quadrants at positions corresponding to each of the 12 hours of the clock. The protocol's 512 × 128 macular cube was used to measure macular thickness.

The same experienced technician performed all the OCTs. Only images with a quality score of at least 7/10 were used. Three consecutive scans of the optic disc and macular region were acquired and analyzed for each eye. The results of the RNFL and GCC measurements were averaged using the data from each of the three scans.

## Pattern electroretinogram

RE-PERG was recorded using a commercial instrument (RETIMAX Advanced ver. 4.3 CSO Florence, Italy) and a method similar to that employed in the PERGLA paradigm, [66] with a few minor changes made by our laboratories. Specifically, we used a stimulus of horizontal bars with a spatial frequency of 1.7 cycles/degree—based on the results of previous studies showing the high sensitivity of this method in detecting RGCs dysfunction in early glaucoma [67,68]—and modulated in counter phase at 15 reversals/s. The stimulus was electronically generated on a high-resolution ionized-gas electrically charged plasma display (contrast: 90% luminance: 80 cd/m$^2$; field size: 24˚ [width] × 24˚ [height]).

The pupils of the patients or NC were 3–4 mm, undilated, and an appropriate correction was made for the working distance (57 cm). The signals were recorded from a 9-mm Ag/AgCl skin electrode placed on the lower eyelid. A similar electrode placed on the lid of the non-stimulated eye was used as a reference, as described in other studies. The impedance was maintained below 5 K. The responses were amplified (gain of 100,000), filtered (bandwidth: 130 Hz) and sampled with a resolution of 12 bits. The analysis time was equal to the period of the stimulus (133 ms).

An average of 650 PERG events (5 consecutive blocks of 130 events) for RE-PERG was calculated, with the automatic rejection of artifacts. The data were then exported to a text file and the mean amplitude (μV) and phase (πrad) of the 2nd harmonic were analyzed by Fourier transform.

The repeatability of the phase of the second harmonic was calculated as the standard deviation of the phase (SDPh). The repeatability of the amplitude (Amp) was not considered, because of a habituation effect. [69] The noise level arising from recording a response to an occluded stimulus was ≤0.087 ± 0.03 μV in both NC and patients. Figs 1–3 show examples of a block of five events in NC and in VD and AD patients. The PERG Amp and PERG SDph values are highlighted. In our laboratory, a PERG Amp value <1.5 μv and PERG SDph values >0.15 SD are considered to indicated pathology. The study was double blind in its design and all RE-PERGs were conducted by the same operator (A. Mavilio).

Statistical analyses were performed using Medcalc® 18.11.3. Because of the high correlation of the responses of the two eyes of the same person, only the data from one randomly chosen eye was included in the analysis. [70]

The distribution of the data was tested for normality using the Shapiro–Wilk test, and a t test was used to determine the differences between two independent groups. Comparisons of more than two independent groups were performed using a one-way ANOVA with post-hoc analyses based on the Scheffe method. The relationships between the electrophysiological values and the SAP, peripapillary RNFL thickness and GCC thickness values were calculated using Pearson's correlation tests. A chi-squared test was used to compare the groups with respect to the categorical variables (sex). A p value < 0.05 was considered to indicate statistical significance.

The Ethics Committee of the Brindisi Social Health District approved the study, and the study protocol adhered to the tenets of the Declaration of Helsinki. Written informed consent was obtained from each participant after administration of the University of California, San Diego Brief Assessment of Capacity to Consent (UBACC). [71]

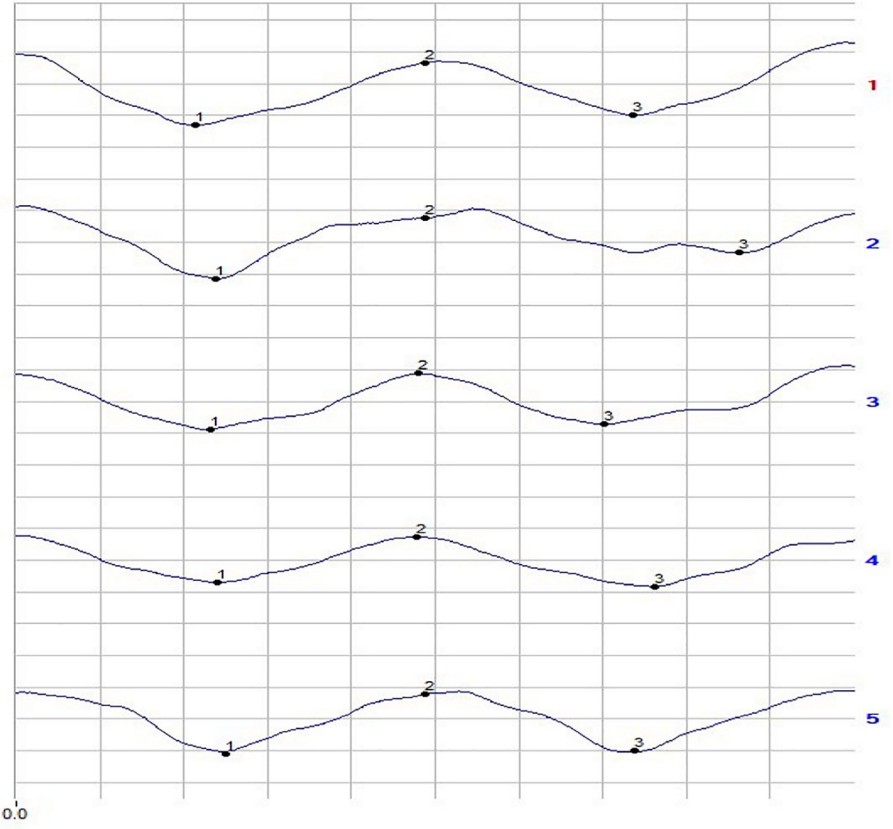

| | |
|---|---|
| Frequency (Hz): | 15.04 |
| amp 1 (uv): | 2.59 |
| Phase 1 (rad): | 0.16 |
| Amp mean (uV): | 2.18 |
| Phase mean (rad): | 0.09 |
| Amp SD (uV): | 0.26 |
| Phase SD (rad): | 0.10 |
| Acq. time (ms): | 133.0 |
| Gain: | 50000 |
| High Pass (Hz): | 1.0 |
| Low Pass (Hz): | 30.0 |
| Events: | 130 |
| Sequences: | 5 |

**Fig 1.**

## Results

The study population consisted of Italians with an education level equal to that of the 8th grade in the USA. All participants lived in Apulia at the time of their enrollment in the study, between 2017 and 2018. For some patients, the family doctor had requested a neuropsychological evaluation for suspected deterioration or dementia, based on cognitive-memory loss reported by the patients; for others, a neurological examination was requested by a neurologist for various reasons, including suspicion of dementia.

Initially, 58 patients were enrolled. However, because of unreliable visual field examinations or poor-quality OCT images, 6 were excluded (4 from the AD group and 2 from the VD group), leaving 52 patients in the study.

The 17 patients in the AD group (5 males and 12 females) ranged in age between 58 and 81 years. Most were retired and came to the visit with a caregiver (usually a family member). Some had active interests, but others did not.

The 16 age-matched patients in the VD group (9 males and 7 females) had not been diagnosed with AD.

The 19 members of the HC group (12 males and 7 females) were also age-matched with the patients. Demographic and other data of the study participants are summarized in Table 1. The results of the statistical analyses are reported in Table 2. There was no difference between groups with respect to age, mean deviation, pattern standard deviation (PSD), RNFL and GCC, as determined in an ANOVA. AD patients had a slight significant reduction in the PERG Amp ($1.33 \pm 0.28$ vs $1.67 \pm 0.16$, $p = 0.01$) value compared to NC whereas the difference

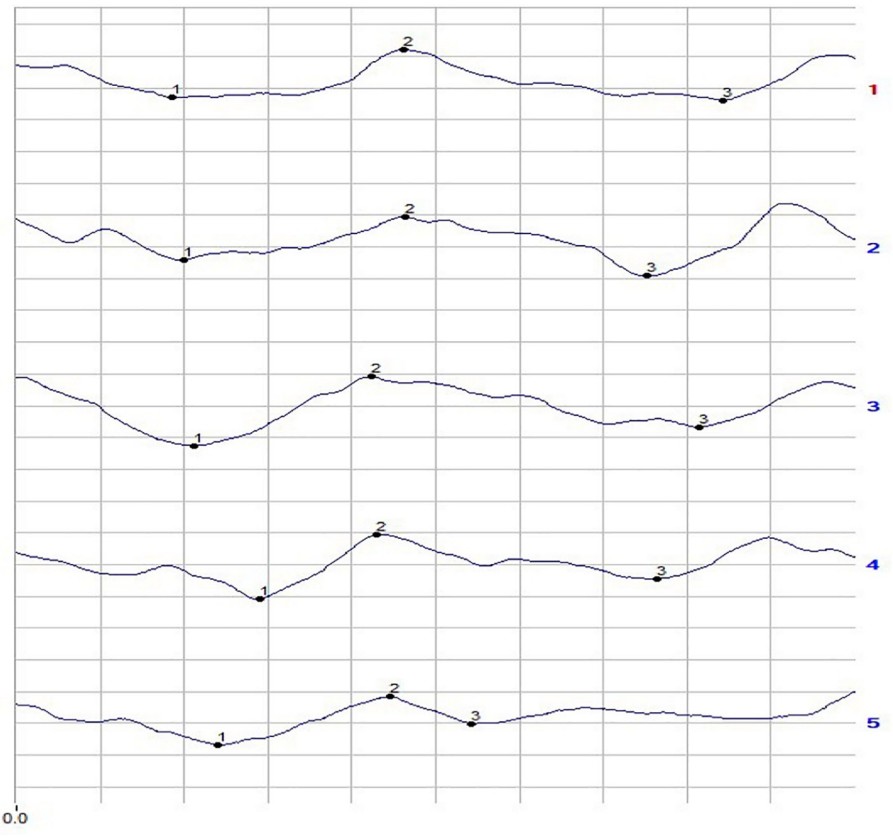

| Frequency (Hz): | 15.04 |
|---|---|
| amp 1 (uv): | 1.58 |
| Phase 1 (rad): | 0.03 |
| Amp mean (uV): | 1.52 |
| Phase mean (rad): | 0.21 |
| Amp SD (uV): | 0.47 |
| Phase SD (rad): | 0.11 |
| Acq. time (ms): | 133.0 |
| Gain: | 50000 |
| High Pass (Hz): | 1.0 |
| Low Pass (Hz): | 30.0 |
| Events: | 130 |
| Sequences: | 5 |

**Fig 2.**

in the PERG SDph was highly significant between AD and VD patients (0.32±0.91 vs 0.12± 0.04, p<0.001) and between AD patients and NC (0.32±0.91 vs 0.12±0.03, p<0.001) (Figs 4 and 5).

The MMSE score was significantly lower in both AD and VD patients than in NC (P = 0.02; P = 0.01 respectively).

The results of the correlation analysis are reported in Table 3.

There was a negative correlation between PERG Amp and age and between MMSE and PSD. Positive correlations were determined for PERG Amp and increasing PERG SDPh, for RNFL thinning and GCC thinning and for a reduction in PERG Amp and RNFL thinning.

## Discussion

The two most frequent causes of dementia worldwide are AD and VD, and their prevalence is expected to increase as populations age. Both diseases may be preceded by MCI, which is common in the elderly population but not necessarily associated with subsequent dementia. AD is associated especially with amnestic MCI, and VD with executive dysfunction and psychomotor slowness, [72] but psychometric evaluation findings alone cannot be used to discriminate VD from AD. Both AD and VD are accompanied by visual disturbances, due primarily to retinal degeneration and retrograde degeneration, respectively. The early diagnosis of AD may allow better disease management, including a delay of symptom occurrence. However, the most accurate tests for the diagnosis of AD are expensive or invasive. Consequently, there is a growing need for the detection of new, less-invasive and more cost-effective diagnostic testing.

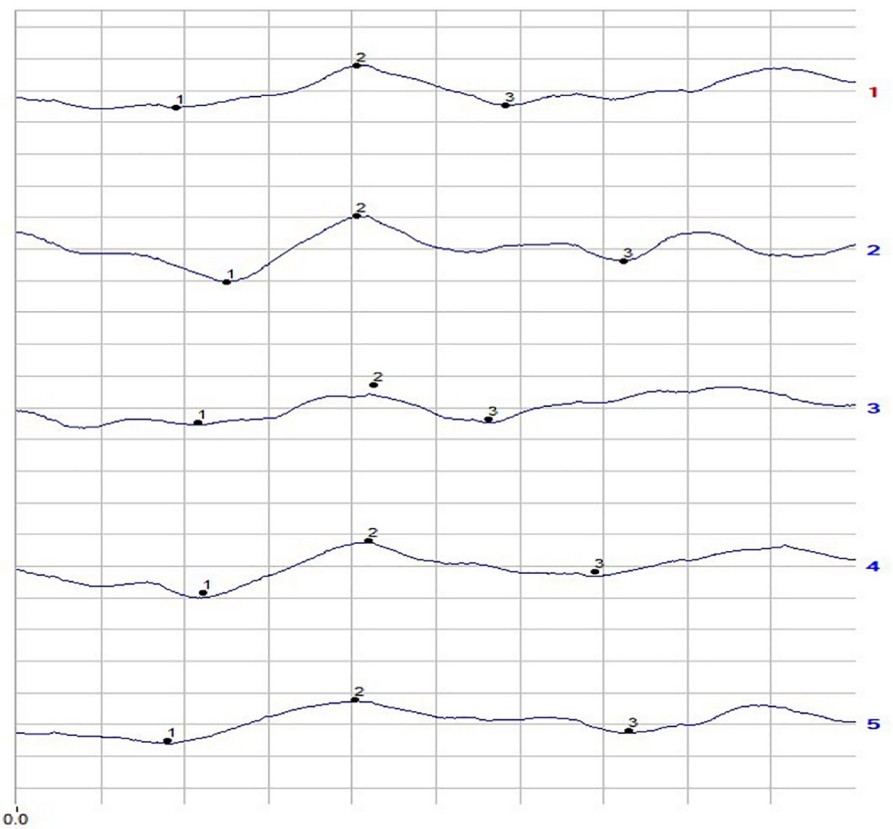

**Fig 3.**

In most AD patients, the visual association cortices are altered whereas the primary visual cortex is spared. [73] Involvement of other areas of the visual pathway is controversial: as in some patients alterations of stereopsis and contrast sensitivity have been reported even in those without evidence of plaques and neurofibrillary tangles. [74] Other studies have demonstrated an involvement of the magnocellular pathway (a visual pathway extending from the inner layers of the retina to the primary visual cortex) in the form of the deposition of a specific type of plaque in the lateral geniculate nucleus as well as in RGCs and their axons. [75] Based on these observations, an evaluation of the macular RGC layer may provide useful diagnostic information for patients with suspected AD. The RGC layer can be studied by imaging and electrophysiological tests, PERG and visual evoked potentials. [23,76,77] For example, ssPERG tests conducted in a mouse model of AD showed alterations in the amplitude of the second wave. [78] However, in ssPERG testing the studied parameter is usually the amplitude, but it can be influenced by causes not related to neurodegeneration, such as optic media opacities and myopia, whereas the phase is not. Thus, we developed a new test, RE-PERG, in which the variability of the phase is studied based on five consecutive ssPERG stimulations. In previous studies we showed that phase variability is higher in glaucoma patients and that it is not influenced by cataract or myopia. [57,58] Since the neurodegeneration of RGCs shows similar features in glaucoma and AD, we examined the ability of RE-PERG to identify early-stage AD and to discriminate AD from VD on the basis of the different mechanisms of neurodegeneration. The results showed a slightly significant reduction in the PERG Amp value in AD patients vs. NC, but no difference between VD patients and NCs. However, the difference in the PERG SDPh in AD vs. VD patients and in AD versus NC patients was highly significant; therefore,

**Table 1. Demographics and specific data.**

| No | Type | gender | age, years | PERG SDPh | PERG Amp (µV) | MD (dB) | PSD (dB) | RNFL (µm) | GCC (µm) | MMSE |
|---|---|---|---|---|---|---|---|---|---|---|
| 1 | VD | m | 68 | 0.07 | 1.09 | -1.69 | 1.54 | 88 | 76 | 25 |
| 2 | VD | m | 57 | 0.12 | 1.89 | -0.5 | 1.3 | 85 | 90 | 24 |
| 3 | VD | m | 75 | 0.1 | 1.55 | -1 | 1 | 94 | 81 | 25 |
| 4 | VD | m | 72 | 0.15 | 1.47 | 1.46 | 1.4 | 90 | 77 | 27 |
| 5 | VD | m | 64 | 0.09 | 1.44 | 1.44 | 0.8 | 85 | 72 | 28 |
| 6 | VD | f | 66 | 0.15 | 1.6 | -0.97 | 1 | 93 | 67 | 28 |
| 7 | VD | f | 76 | 0.1 | 1.72 | 0.63 | 1.2 | 77 | 73 | 27 |
| 8 | VD | m | 68 | 0.15 | 1.77 | -0.97 | 1 | 93 | 67 | 27 |
| 9 | VD | f | 80 | 0.11 | 1.42 | 1.01 | 1.4 | 95 | 92 | 26 |
| 10 | VD | f | 62 | 0.06 | 1.55 | 0.77 | 0.74 | 96 | 82 | 28 |
| 11 | VD | f | 67 | 0.23 | 1.53 | 0.85 | 1.02 | 92 | 80 | 27 |
| 12 | VD | m | 84 | 0.17 | 1.42 | 0.93 | 1.24 | 81 | 79 | 26 |
| 13 | VD | m | 75 | 0.12 | 1.55 | 0.47 | 0.94 | 90 | 77 | 26 |
| 14 | VD | m | 79 | 0.14 | 1.75 | 1.01 | 0.87 | 94 | 80 | 18 |
| 15 | VD | f | 72 | 0.14 | 1.7 | 0.88 | 1.5 | 94 | 90 | 22 |
| 16 | VD | f | 67 | 0.05 | 2.1 | -0.04 | 2.2 | 99 | 81 | 18 |
| 17 | AD | f | 69 | 0.2 | 1.21 | 0.04 | 1.17 | 93 | 77 | 19 |
| 18 | AD | m | 81 | 0.61 | 1 | -0.43 | 1.47 | 87 | 72 | 27 |
| 19 | AD | f | 74 | 0.15 | 1.34 | 0.89 | 1.34 | 84 | 77 | 24 |
| 20 | AD | f | 60 | 0.07 | 1.5 | -1.29 | 2.05 | 98 | 84 | 23 |
| 21 | AD | f | 81 | 0.14 | 1.28 | -0.78 | 1.28 | 81 | 77 | 28 |
| 22 | AD | m | 66 | 0.48 | 1.16 | -0.68 | 1.9 | 76 | 74 | 24 |
| 23 | AD | f | 70 | 0.22 | 1.67 | 0.47 | 1.37 | 97 | 82 | 24 |
| 24 | AD | f | 58 | 0.18 | 1.82 | 0.71 | 1.39 | 106 | 92 | 24 |
| 25 | AD | f | 78 | 0.39 | 0.93 | 1.51 | 1.43 | 88 | 80 | 24 |
| 26 | AD | m | 81 | 0.47 | 1.57 | 1.51 | 1.43 | 93 | 72 | 23 |
| 27 | AD | f | 67 | 0.11 | 1.57 | 0.23 | 1.8 | 81 | 77 | 21 |
| 28 | AD | m | 72 | 0.25 | 1.32 | 0.5 | 1.8 | 70 | 55 | 22 |
| 29 | AD | m | 77 | 0.66 | 0.84 | -0.75 | 1.22 | 82 | 76 | 25 |
| 30 | AD | f | 79 | 0.58 | 1.11 | 1.34 | 0.97 | 80 | 65 | 28 |
| 31 | AD | f | 60 | 0.3 | 1.1 | 0.8 | 1.12 | 77 | 75 | 29 |
| 32 | AD | f | 76 | 0.14 | 1.52 | -0.33 | 1.12 | 80 | 81 | 28 |
| 33 | AD | f | 70 | 0.46 | 1.66 | 1.15 | 1.1 | 87 | 80 | 28 |
| 34 | NC | m | 74 | 0.09 | 1.62 | -0.4 | 0.8 | 88 | 72 | 28 |
| 35 | NC | m | 74 | 0.13 | 1.78 | -0.5 | 1.3 | 74 | 64 | 27 |
| 36 | NC | m | 78 | 0.1 | 1.46 | -1.01 | 1.2 | 88 | 70 | 29 |
| 37 | NC | m | 68 | 0.12 | 1.65 | 2 | 1.5 | 104 | 91 | 27 |
| 38 | NC | m | 70 | 0.11 | 1.58 | 1.3 | 1.55 | 101 | 91 | 26 |
| 39 | NC | f | 74 | 0.12 | 1.79 | 1.1 | 0.88 | 87 | 71 | 27 |
| 40 | NC | m | 70 | 0.1 | 1.71 | -0.97 | 1 | 95 | 76 | 28 |
| 41 | NC | m | 65 | 0.1 | 1.5 | 0.63 | 1.2 | 96 | 83 | 27 |
| 42 | NC | f | 60 | 0.1 | 1.81 | 1.01 | 0.87 | 80 | 80 | 25 |
| 43 | NC | f | 64 | 0.08 | 1.59 | 0.85 | 1.02 | 93 | 87 | 26 |
| 44 | NC | m | 65 | 0.1 | 2.14 | -0.23 | 1.1 | 79 | 70 | 28 |
| 45 | NC | m | 60 | 0.11 | 1.51 | -2.1 | 1.53 | 99 | 90 | 28 |
| 46 | NC | m | 68 | 0.16 | 1.53 | 0.93 | 1.46 | 84 | 80 | 29 |
| 47 | NC | m | 66 | 0.1 | 1.69 | 1.01 | 1.4 | 88 | 74 | 30 |
| 48 | NC | m | 77 | 0.16 | 1.75 | 1.81 | 1.41 | 82 | 80 | 28 |

*(Continued)*

**Table 1.** (Continued)

| No | Type | gender | age, years | PERG SDPh | PERG Amp (µV) | MD (dB) | PSD (dB) | RNFL (µm) | GCC (µm) | MMSE |
|----|------|--------|-----------|-----------|---------------|---------|----------|-----------|----------|------|
| 49 | NC | f | 86 | 0.1 | 1.65 | -0.5 | 1.3 | 92 | 86 | 25 |
| 50 | NC | f | 62 | 0.15 | 1.5 | -0.45 | 1.34 | 70 | 87 | 25 |
| 51 | NC | m | 66 | 0.2 | 1.75 | 1 | 1.4 | 100 | 90 | 27 |
| 52 | NC | m | 66 | 0.17 | 1.76 | 0.4 | 1.5 | 99 | 90 | 25 |

Mean Deviation (MD) Pattern Standard Deviation (PSD), Retinal Nerve Fiber Layer Thickness (RNFL), ganglion cell complex (GCC), steady-state intrinsic phase variability (PERG SDph) steady-state PERG amplitude (PERG Amp), Mini-Mental State Examination (MMSE) in Early Alzheimer disease (AD), Vascular Dementia-related MCI (VD) and Normal Controls (NC)

PERG SDPh may be of value not only in detecting inner retinal dysfunction in AD, but also in distinguishing between AD and VD.

Correlation studies showed a negative correlation between PERG Amp and age, as expected due to the physiological loss of RGCs. The negative correlation between MMSE and PSD, that is, a worsening of the visual field related to a reduction in the MMSE score, may reflect the neurodegeneration occurring both in the retina and in the brain. The positive correlation between PERG Amp reduction and an increased PERG SDPh can be explained by a worsening of all parameters with disease progression, and that between RNFL and RGC thinning by the parallel degeneration of neuronal cell bodies and axons (Table 3). The positive correlation between PERG Amp reduction and RNFL thinning indicates that the amplitude is related to the number of surviving RGCs.The findings of our study suggest that PERG SDph is a suitable parameter to detect early damage to magnocellular RGCs in AD patients. While the M system has been shown to respond to stimuli of high temporal frequency, a response by the P system cannot be excluded, also because the K and P visual streams were not specifically tested. However, there are fewer P cells and they tend to be scattered, such that the increased PERG SDph could be predominantly attributed to M dysfunction. Our finding is in agreement with other studies in which involvement of the M pathway was reported. [23]

As noted above, the phase variation is related to the synaptic loss and dendritic degeneration that may precede ganglion cell loss. [47] Such alterations have been described in early AD, but also in Parkinson's and Huntington's diseases. [79] Normal neuronal activity is

**Table 2. Demographic and relevant ocular characteristic of study participants.**

|  | AD (17) | | VD (16) | | NC (19) | | P-value[®] | | |
|--|---------|--|---------|--|---------|--|-----------|--|--|
|  | Mean | SD | Mean | SD | Mean | SD | AD vs VD | VD vs NC | AD vs NC |
| age | 71.71 | 7.63 | 70.75 | 7.17 | 69.10 | 6.67 | P = 0.72 | P = 0.5 | P = 0.44 |
| PERG Amp (µv) | 1.33 | 0.28 | 1.59 | 0.23 | 1.67 | 0.16 | P = 0.2 | P = 0.26 | P = 0.01 |
| PERG SDph | 0.32 | 0.19 | 0.12 | 0.04 | 0.12 | 0.03 | P<0.001 | P = 0.95 | P<0.001 |
| MD (db) | 0.29 | 0.88 | 0.27 | 0.99 | 0.31 | 1.07 | P = 0.5 | P = 0.90 | P = 0.31 |
| PSD (db) | 1.41 | 0.31 | 1.20 | 0.37 | 1.25 | 0.24 | P = 0.6 | P = 0.24 | P = 0.52 |
| GCC (µm) | 76.23 | 7.96 | 79.00 | 7.39 | 80.63 | 8.60 | P = 0.6 | P = 0.24 | P = 0.23 |
| RNFL (µm) | 85.89 | 9.18 | 90.38 | 5.85 | 89.42 | 9.56 | P = 0.1 | P = 0.73 | P = 0.77 |
| MMSE | 24.76 | 2.84 | 25.12 | 3.20 | 27.10 | 1.49 | P = 0.73 | P = 0.02 | P = 0.01 |

Mean Deviation (MD) Pattern Standard Deviation (PSD), Retinal Nerve Fiber Layer Thickness (RNFL), ganglion cell complex (GCC), steady-state intrinsic phase variability (PERG SDph) steady-state PERG amplitude (PERG Amp), Mini-Mental State Examination (MMSE) in Early Alzheimer disease (AD), Vascular Dementia-related MCI (VD) and Normal Controls (NC)

*–One Way Analysis of Variance (Bonferroni corrected); **—Chi-Square

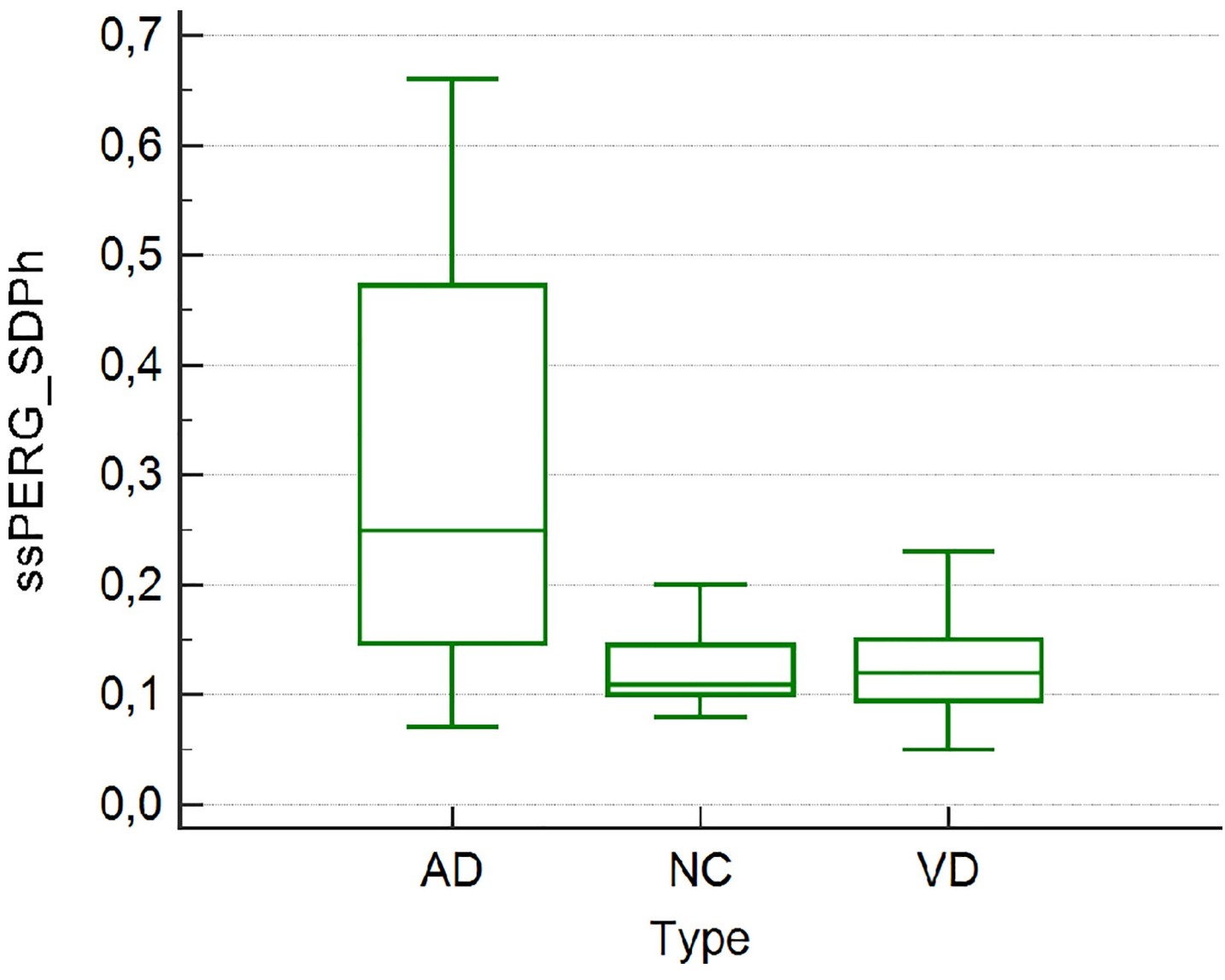

**Fig 4.**

accompanied by a high energy demand; [80] such that RE-PERG serves as a metabolic stress test able to show early damage to RGCs.

In glaucoma patients, functional and anatomical changes may be present in RGCs before any damage of the optic nerve is detectable. [52] Thus, in CNS diseases that share some features of the degeneration seen in glaucoma, the same may be true.

Our study may have been biased by several factors. First, the diagnosis of MCI was based exclusively on the MMSE, which cannot replace a full psychometric evaluation. Tests specific for AD include the Alzheimer's Disease Assessment Scale (ADAS-Cog), the Clinical Dementia Rating (CDR) score, and the Repeatable Battery for the Assessment of Neuropsycological Status (RBANS). In the diagnosis of VD, the Montreal Cognitive Assessment (MoCA) has a higher sensitivity and specificity than the MMSE. [81] However, our study was performed in a National Health Service setting, and MMSE is the only psychometric test available. Thus, it cannot be ruled out that a more specific evaluation would have led to a different definition of mental status and influenced our results.

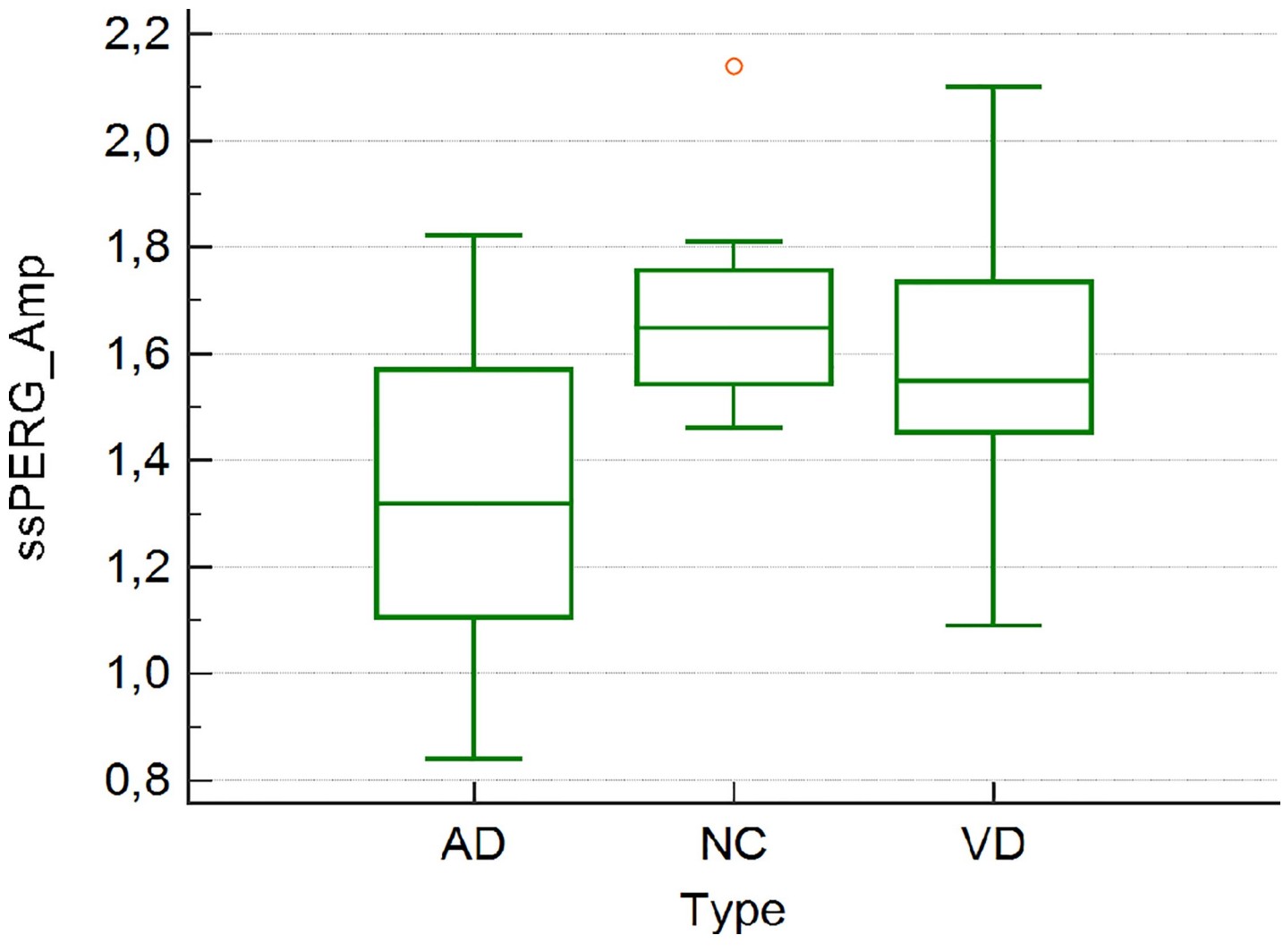

**Fig 5.**

Another possible source of bias was the small number of enrolled patients. Further studies with a larger cohort of patients are required to confirm our preliminary results.

Two issues emerge from this study. The first is the question whether the alteration in PERG SDPh is a sign of primary RGCs degeneration or related to transsynaptic degeneration in the visual cortex. In our opinion, the first hypothesis is more likely, as RGC thinning has been found both in prodromal and in preclinical AD as well as in patients without other signs of visual cortex involvement. [82,83]

The second is the shared finding of an altered PERG SDph in both glaucoma and AD. AD and glaucoma have several common features. Epidemiological studies have shown that the prevalence of glaucoma in AD patients is about 25% vs. 5–6% in the non-AD population. [84,85] Abnormal folded amyloid beta (Aβ) and tau protein, typical findings in AD, have been demonstrated both in mouse models of glaucoma and in humans with the disease. [86,87] In addition, several studies have shown OCT alterations typical of glaucoma, such as RNFL and RGC thinning, in patients with early and even preclinical AD, [88,89] and visual field alterations detected in glaucoma, including arcuate defects, also occur in AD. [90] Finally, an

**Table 3. Correlation table.**

|  |  | age | MD (dB) | PSD (dB) | GCC (µm) | RNFL (µm) | PERG Amp (µV) | PERG SDPh | MMSE |
|---|---|---|---|---|---|---|---|---|---|
| age | CC |  | 0.12 | -0.09 | -0.27 | -0.17 | -0.29 | 0.3 | -0.02 |
|  | SL-P |  | 0.4 | 0.5 | 0.05 | 0.22 | 0.03 | 0.03 | 0.90 |
| MD (db) | CC | 0.12 |  | -0.13 | 0.13 | 0.06 | 0.09 | 0.11 | 0.002 |
|  | SL-P | 0.4 |  | 0.36 | 0.35 | 0.67 | 0.53 | 0.43 | 0.99 |
| PSD (db) | CC | -0.09 | -0.13 |  | 0.17 | 0.06 | -0.07 | 0.07 | -0.42 |
|  | SL-P | 0.54 | 0.36 |  | 0.22 | 0.69 | 0.64 | 0.64 | 0.0018 |
| GCC (µm) | CC | -0.27 | 0.13 | 0.17 |  | 0.59 | 0.21 | -0.27 | -0.14 |
|  | SL-P | 0.051 | 0.35 | 0.22 |  | <0.0001 | 0.13 | 0.054 | 0.31 |
| RNFL (µm) | CC | -0.17 | 0.06 | 0.06 | 0.59 |  | 0.28 | -0.24 | -0.15 |
|  | SL-P | 0.22 | 0.67 | 0.69 | <0.0001 |  | 0.04 | 0.08 | 0.29 |
| PERG Amp (µv) | CC | -0.29 | 0.09 | -0.07 | 0.21 | 0.28 |  | -0.6 | -0.05 |
|  | SL-P | 0.038 | 0.52 | 0.63 | 0.13 | 0.05 |  | <0.0001 | 0.71 |
| PERG SDPh | CC | 0.3 | 0.11 | 0.07 | -0.27 | -0.24 | -0.6 |  | -0.006 |
|  | SL-P | 0.03 | 0.43 | 0.64 | 0.054 | 0.08 | <0.0001 |  | 0.97 |
| MMSE | CC | -0.02 | 0.002 | -0.42 | -0.14 | -0.15 | -0.05 | -0.006 |  |
|  | SL-P | 0.91 | 0.99 | 0.0018 | 0.31 | 0.29 | 0.71 | 0.97 |  |

Pearson Correlation Coefficient (CC) and Significance Level P (SL-P) between Mean Deviation (MD) Pattern Standard Deviation (PSD), Retinal Nerve Fiber Layer Thickness (RNFL), ganglion cell complex (GCC), steady-state intrinsic phase variability (PERG SDph) steady-state PERG amplitude (PERG Amp) and Mini-Mental State Examination (MMSE) in all participants

enlarged cup-to-disc ratio of the optic nerve, the most typical feature of glaucoma, has also been detected in some, [91–93] but not all [94,95] AD patients.

Recently, optical coherence tomography angiography (OCT-A) has been used to study AD. Bulut et al. reported a lower retinal vascular density (VD) and choroidal thickness [96] together with an enlargement of the foveal avascular zone (FAZ) in AD patients compared to NC. In a comparison of AD and primary open-angle glaucoma (POAG) patients, Zabel et al. found a larger FAZ and a reduced vascular density in the deep vascular plexus in the AD group [97] whereas in POAG patients reductions in the vascular density of the superficial vascular plexus and in radial peripapillary capillaries were detected. However, a reduced VD and FAZ enlargement have also been reported in normal-tension glaucoma (NTG). [98] In addition, an even larger FAZ occurs in progressed glaucoma (both NTG and POAG) [99] and in POAG patients with central visual field defects. [100] The FAZ is also variably influenced by glaucoma surgery. [101]

Finally, the reduced VD of the deep macular plexus, such as reported by Zabel et al. in AD patients, is also a feature of progressed NTG. [99] Thus, whether OCT-A findings comprise a specific biomarker of AD remains to be determined in further studies. Moreover, these studies also demonstrate that all of the tools used to diagnose glaucoma may be biased by the presence of AD. In the absence of an elevated intraocular pressure, i.e. in a patient with NTG, the differential diagnosis can be particularly challenging and AD has to be carefully ruled out.

Other causes of inner retinal dysfunction, detectable by electrophysiological tests, as stated before, are Multiple Sclerosis and Parkinson's disease; we didn't test RE-PERG in these diseases, but its alteration cannot be excluded. At the same way, it is also known that age-related visual conditions such as age itself, presbyopia and cataract can influence PERG. As for cataract, we showed reduced amplitude with small intrinsic variability of the phase in a RE-PERG pilot study,[57] but further studies are required also in the other above-mentioned conditions. Our results suggest that RE-PERG is a quick, easy to perform and non-invasive test able to

detect RGC dysfunction in AD, but despite its promise its utility must be confirmed in other laboratories and in larger cohorts of patients.

## Author Contributions

**Conceptualization:** Alberto Mavilio.

**Data curation:** Alberto Mavilio.

**Funding acquisition:** Alberto Mavilio.

**Investigation:** Alberto Mavilio, Florenza Prete, Viviana Guadalupi, Rosanna Dammacco.

**Methodology:** Alberto Mavilio.

**Software:** Alberto Mavilio.

**Validation:** Alberto Mavilio.

**Visualization:** Giovanni Alessio.

**Writing – review & editing:** Dario Sisto.

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
