## [Decision Letter · Decision Letter 0]

31 Dec 2019

PONE-D-19-31892

RE-PERG in early-onset Alzheimer’s Disease. A double-blind, electrophysiological pilot study

PLOS ONE

Dear Dr. Mavilio,

Thank you for submitting your manuscript to PLOS ONE. After careful consideration by 2 Reviewers and an Academic Editor, all of the critiques of both Reviewers must be addressed in detail in a revision to determine publication status. If you are prepared to undertake the work required, I would be pleased to reconsider my decision, but revision of the original submission without directly addressing the critiques of the two Reviewers does not guarantee acceptance for publication in PLOS ONE. If the authors do not feel that the queries can be addressed, please consider submitting to another publication medium. A revised submission will be sent out for re-review. The authors are urged to have the manuscript given a hard copyedit for syntax and grammar.

**Comments to the Author**

1. Is the manuscript technically sound, and do the data support the conclusions?

Reviewer #1: Yes

Reviewer #2: Yes

2. Has the statistical analysis been performed appropriately and rigorously? 

Reviewer #1: Yes

Reviewer #2: Yes

3. Have the authors made all data underlying the findings in their manuscript fully available?

Reviewer #1: Yes

Reviewer #2: Yes

4. Is the manuscript presented in an intelligible fashion and written in standard English?

Reviewer #1: Yes

Reviewer #2: Yes

5. Review Comments to the Author

Reviewer #1: Introduction:

1. Authors should explain in more detail what Alzheimer's disease is and how it differs from (VD) -related MCI, describe changes in the CNS, retina, and why eye biomarkers may be useful.

2. How do we confirm the diagnosis of AD in sMRI?

MATERIALS AND METHODS:

1. The authors described that neurologic exclusion criteria were: „neurological / psychiatric conditions other than mild AD"- what about patients who had vascular dementia? After all, they were included in the study.

2. Did all participants have an MRI? How on the basis of sMRI diagnosed and distinguish patients with AD and VD.

3. The description of the statistics should be more detailed, paying attention to the tests used.

Results:

1. „ANOVA analysis showed no difference between groups for age. MD. PSD RNFL and GCC.”- errors in punctuation

2. Please specify the number and characteristics of excluded subjects to reduce the selection bias.

Discussion:

1. Epidemiology, diagnosis and pathogenesis should be included in the introduction (not in the discussion).

2. Unfortunately, the most effective tests for the diagnosis of AD are expensive and invasive.”- e.g. MRI is not an invasive test, therefore this sentence should be formulated more precisely

3. "Since the neurodegeneration of RGCs shows similar features in both glaucoma and AD, we performed this study in order to evaluate the ability of RE-PERG in the identification of the early stage of AD. "- How can RE-PERG distinguish AD from glaucoma? Since the PERG test is used in glaucoma diagnosis why the authors did not compare glaucoma, AD and HC?

4. The authors did not describe the limitations of their research

5. MMSE may not be the best cognitive test with which to measure AD related cognitive impairment. This should be acknowledged and the limitations explored in the discussion.

6. Why do the authors compare patients with VD to AD and NC if the problem is to distinguish AD from glaucoma? I think that vascular changes in the retina in AD patients should be mentioned (Bulut, M., Kurtuluş, F., Gözkaya, O., Erol, M. K., Cengiz, A., Akıdan, M., & Yaman, A. (2018). Evaluation of optical coherence tomography angiographic findings in Alzheimer’s type dementia. British Journal of Ophthalmology, 102(2), 233-237.) and describe differences in microvascularization between AD, POAG and NC (Zabel, P., Kaluzny, J. J., Wilkosc-Debczynska, M., Gebska-Toloczko, M., Suwala, K., Zabel, K., ... & Araszkiewicz, A. (2019). Comparison of Retinal Microvasculature in Patients With Alzheimer's Disease and Primary Open-Angle Glaucoma by Optical Coherence Tomography Angiography. Investigative ophthalmology & visual science, 60(10), 3447-3455.)

References:

1. Kamila K., Wojciech L., Andrzej P. Pattern electroretinogram (PERG) and pattern visual evoked potential (PVEP) in the early stages of Alzheimer’s disease Doc Ophthalmol. 2010 435 Oct; 121(2): 111–121.”- these are not the author's' last names

Reviewer #2: Manuscript n.: PONE-D-19-31892

TITLE: RE-PERG in early-onset Alzheimer’s disease. A double-blind electrophysiological pilot study.

Article Type: Research Article.

In this study, the Authors investigate the ability RE-PERG (retest PERG) in detection inner retinal bioelectric function abnormalities in patients with early-onset Alzheimer Disease (AD). They investigated a series of pt. (17 pt.) with AD and report mean RE-PERG amplitude significantly lower and phase of 2nd harmonic (PERG SDPh) higher in AD pts. compared with controls. Moreover, they affirm that RE-PERG can be useful to identify early AD stages and that changes may be imputable to magnocellular pathway dysfunction not present in other conditions, like vascular dementia and conclude that RE-PERG could be a new promising biomarker of neurodegenerative disease.

The study is well conducted and written, results are clear, sound well and could be interesting for the readership of the journal, even if not at all new (see previous paper on the same field reported in references).

The manuscript has some issues needed to be addressed.

Major criticism are:

- First, the AA address the magnocellular pathway (M pathway), but it should be kept in mind that there are other pathway subsystems (Parvo-, P and Konio-cellular, K subsystem; see Paper of Livingstone, Porciatti, van Essen etc. on this topic). The AA should add some information concerning these subsystems and their propriety. In fact whereas color information is processed mainly by the P system, luminance by both P and M subsystem. Moreover, the stimulus employed by the AA is not selective for the Magno and result could be aspecific. The AA should add some informations on visual pathways subsystem and property.

- Second the AA to exclude a bias in their study should exclude involvement of other subsystem before to affirm an exclusive magnocellular dysfunction in their conclusions. Please clarify.

- Third, How can the AA exclude that a lower RE-PERG and a higher SDPh RE-PERG are not an aspecific changes

Keywords: I suggest add visual pathway subsystem.

Abstract: modify according the above criticisms.

Introduction: see major criticisms.

In line 89 they affirm that “ … this parameter is not influenced by optical …”; I suggest “scarcely influenced by refraction “

Materials and methods: -

Discussion and Concluding remarks: they have to be rewritten following the above suggestions and comments.

References: there are only few typing mistaken and two reference in my opinion seems the same (n. 14 and 56): check.

6. PLOS authors have the option to publish the peer review history of their article (what does this mean?). If published, this will include your full peer review and any attached files.

**Do you want your identity to be public for this peer review?** For information about this choice, including consent withdrawal, please see our Privacy Policy.

Reviewer #1: Yes: Przemyslaw Zabel

Reviewer #2: Yes: Ferdinando Sartucci

We would appreciate receiving your revised manuscript by June, 2020. To enhance the reproducibility of your results, we recommend that if applicable you deposit your laboratory protocols in protocols.io, where a protocol can be assigned its own identifier (DOI) such that it can be cited independently in the future. For instructions see: http://journals.plos.org/plosone/s/submission-guidelines#loc-laboratory-protocols

We look forward to receiving your revised manuscript.

Kind regards,

Stephen D. Ginsberg, Ph.D.

Section Editor

PLOS ONE

2) Please describe in your methods section how capacity to provide consent was determined for the participants in this study. Please also state whether your ethics committee or IRB approved this consent procedure. If you did not assess capacity to consent please briefly outline why this was not necessary in this case.

3) Thank you for including your ethics statement:

"The Institutional Review Board and Ethics Committee of the institute approved the study, and the study protocol adhered to the tenets of the Declaration of Helsinki. Written informed consent was obtained from each participating patient."

i) Please amend your current ethics statement to include the full name of the ethics committee/institutional review board(s) that approved your specific study.

ii) Once you have amended this/these statement(s) in the Methods section of the manuscript, please add the same text to the “Ethics Statement” field of the submission form (via “Edit Submission”).

4) Thank you for stating the following financial disclosure:

 [NO].

Please provide an amended Funding Statement that declares *all* the funding or sources of support received during this specific study (whether external or internal to your organization) as detailed online in our guide for authors at http://journals.plos.org/plosone/s/submit-now.  

Please state what role the funders took in the study.  If any authors received a salary from any of your funders, please state which authors and which funder. If the funders had no role, please state: "The funders had no role in study design, data collection and analysis, decision to publish, or preparation of the manuscript."

5) Thank you for stating the following in your Competing Interests section: 

[NO].

i) Please complete your Competing Interests on the online submission form to state any Competing Interests. If you have no competing interests, please state "The authors have declared that no competing interests exist.", as detailed online in our guide for authors at http://journals.plos.org/plosone/s/submit-now

ii)  This information should be included in your cover letter; we will change the online submission form on your behalf.

6) Please include your tables as part of your main manuscript and remove the individual files. Please note that supplementary tables (should remain/ be uploaded) as separate "supporting information" files.

---

## [Author Response · Author response to Decision Letter 0]

18 May 2020

Dear Editor,

Enclosed you find the updated version of our manuscript, which has been widely revised according to your reviewers' criticisms. The manuscript has also been corrected by an English medical editor.

The Authors have declared that no competing interest exist.

The Authors received no specific funding for this work

Best regards

5. Review Comments to the Author

Reviewer #1: Introduction:

1. Authors should explain in more detail what Alzheimer's disease is and how it differs from (VD) -related MCI, describe changes in the CNS, retina, and why eye biomarkers may be useful.

 DONE: please, see new introduction

2. How do we confirm the diagnosis of AD in sMRI?

DONE: please, see introduction, lines 93-95

MATERIALS AND METHODS:

1. The authors described that neurologic exclusion criteria were: „neurological / psychiatric conditions other than mild AD"- what about patients who had vascular dementia? After all, they were included in the study.

DONE: line 149 

2. Did all participants have an MRI? How on the basis of sMRI diagnosed and distinguish patients with AD and VD.

DONE: please, see introduction and lines 165-166

3. The description of the statistics should be more detailed, paying attention to the tests used.

DONE: lines 219-228

Results:

1. „ANOVA analysis showed no difference between groups for age. MD. PSD RNFL and GCC.”- errors in punctuation

DONE: lines 249-251

2. Please specify the number and characteristics of excluded subjects to reduce the selection bias.

DONE: lines 239-241

Discussion:

1. Epidemiology, diagnosis and pathogenesis should be included in the introduction (not in the discussion).

DONE: please, see new introduction

2. Unfortunately, the most effective tests for the diagnosis of AD are expensive and invasive.”- e.g. MRI is not an invasive test, therefore this sentence should be formulated more precisely

DONE: please, see lines 269-270

3. "Since the neurodegeneration of RGCs shows similar features in both glaucoma and AD, we performed this study in order to evaluate the ability of RE-PERG in the identification of the early stage of AD. "- How can RE-PERG distinguish AD from glaucoma? Since the PERG test is used in glaucoma diagnosis why the authors did not compare glaucoma, AD and HC?

This is not the aim of our study. This matter is developed in the discussion section; currently no diagnostic test used for glaucoma (i.e. OCT of the optic nerve, visual field examination, electrophysiological tests) can distinguish between glaucoma and AD, this is true also for REPERG. According to some works the only difference seems to be the appareance of the optic nerve (which should show cupping in glaucoma but not in AD) even if according to other authors a similar cupping can be observed also in AD. As we state in the discussion section, this situation can make it difficult to distinguish between glaucoma and AD, especially when a raise of intraocular pressure is not detectable, that is, in case of normal-tension glaucoma suspect.

4. The authors did not describe the limitations of their research

DONE: please, see lines 317-325

5. MMSE may not be the best cognitive test with which to measure AD related cognitive impairment. This should be acknowledged and the limitations explored in the discussion.

DONE: please, see lines 317-325

6. Why do the authors compare patients with VD to AD and NC if the problem is to distinguish AD from glaucoma? I think that vascular changes in the retina in AD patients should be mentioned (Bulut, M., Kurtuluş, F., Gözkaya, O., Erol, M. K., Cengiz, A., Akıdan, M., & Yaman, A. (2018). Evaluation of optical coherence tomography angiographic findings in Alzheimer’s type dementia. British Journal of Ophthalmology, 102(2), 233-237.) and describe differences in microvascularization between AD, POAG and NC (Zabel, P., Kaluzny, J. J., Wilkosc-Debczynska, M., Gebska-Toloczko, M., Suwala, K., Zabel, K., ... & Araszkiewicz, A. (2019). Comparison of Retinal Microvasculature in Patients With Alzheimer's Disease and Primary Open-Angle Glaucoma by Optical Coherence Tomography Angiography. Investigative ophthalmology & visual science, 60(10), 3447-3455.)

DONE: please see our answer to point 3. As for OCT-A, see lines 342-351

References:

1. Kamila K., Wojciech L., Andrzej P. Pattern electroretinogram (PERG) and pattern visual evoked potential (PVEP) in the early stages of Alzheimer’s disease Doc Ophthalmol. 2010 435 Oct; 121(2): 111–121.”- these are not the author's' last names

DONE

Reviewer #2: Manuscript n.: PONE-D-19-31892

TITLE: RE-PERG in early-onset Alzheimer’s disease. A double-blind electrophysiological pilot study.

Article Type: Research Article.

In this study, the Authors investigate the ability RE-PERG (retest PERG) in detection inner retinal bioelectric function abnormalities in patients with early-onset Alzheimer Disease (AD). They investigated a series of pt. (17 pt.) with AD and report mean RE-PERG amplitude significantly lower and phase of 2nd harmonic (PERG SDPh) higher in AD pts. compared with controls. Moreover, they affirm that RE-PERG can be useful to identify early AD stages and that changes may be imputable to magnocellular pathway dysfunction not present in other conditions, like vascular dementia and conclude that RE-PERG could be a new promising biomarker of neurodegenerative disease.

The study is well conducted and written, results are clear, sound well and could be interesting for the readership of the journal, even if not at all new (see previous paper on the same field reported in references).

The manuscript has some issues needed to be addressed.

Major criticism are:

- First, the AA address the magnocellular pathway (M pathway), but it should be kept in mind that there are other pathway subsystems (Parvo-, P and Konio-cellular, K subsystem; see Paper of Livingstone, Porciatti, van Essen etc. on this topic). The AA should add some information concerning these subsystems and their propriety. In fact whereas color information is processed mainly by the P system, luminance by both P and M subsystem. Moreover, the stimulus employed by the AA is not selective for the Magno and result could be aspecific. The AA should add some informations on visual pathways subsystem and property.

DONE: please, see new introduction (lines 71-83)

- Second the AA to exclude a bias in their study should exclude involvement of other subsystem before to affirm an exclusive magnocellular dysfunction in their conclusions. Please clarify.

Please please, see new introduction and comments (lines 77-80)

- Third, How can the AA exclude that a lower RE-PERG and a higher SDPh RE-PERG are not an aspecific changes

By means of exclusion criteria

Keywords: I suggest add visual pathway subsystem.

DONE

Abstract: modify according the above criticisms.

DONE: please, see lines 317-325

Introduction: see major criticisms.

In line 89 they affirm that “ … this parameter is not influenced by optical …”; I suggest “scarcely influenced by refraction “

Our statement is based on exclusion criteria and on our previous works (ref 56-58)

Materials and methods: -

Discussion and Concluding remarks: they have to be rewritten following the above suggestions and comments.

DONE

References: there are only few typing mistaken and two reference in my opinion seems the same (n. 14 and 56): check.

DONE

---

## [Decision Letter · Decision Letter 1]

22 Jun 2020

PONE-D-19-31892R1

RE-PERG in early-onset Alzheimer’s disease:  A double-blind, electrophysiological pilot study

PLOS ONE

Dear Dr. Mavilio,

Thank you for resubmitting your work to PLOS ONE. Please make the corrections posed by Reviewer #2 so I can render a decision on this manuscript.

**Comments to the Author**

1. If the authors have adequately addressed your comments raised in a previous round of review and you feel that this manuscript is now acceptable for publication, you may indicate that here to bypass the “Comments to the Author” section, enter your conflict of interest statement in the “Confidential to Editor” section, and submit your "Accept" recommendation.

Reviewer #1: All comments have been addressed

Reviewer #2: (No Response)

2. Is the manuscript technically sound, and do the data support the conclusions?

Reviewer #1: Yes

Reviewer #2: Yes

3. Has the statistical analysis been performed appropriately and rigorously? 

Reviewer #1: Yes

Reviewer #2: Yes

4. Have the authors made all data underlying the findings in their manuscript fully available?

Reviewer #1: Yes

Reviewer #2: Yes

5. Is the manuscript presented in an intelligible fashion and written in standard English?

Reviewer #1: Yes

Reviewer #2: Yes

6. Review Comments to the Author

Reviewer #1: All the concerns have been addressed. The authors corrected the manuscript in accordance with the suggestions of the reviewers, which made the manuscript suitable for publication.

Reviewer #2: Manuscript n.: PONE-D-19-31892

TITLE: RE-PERG in early-onset Alzheimer’s disease. A double-blind electrophysiological pilot study.

Article Type: Research Article.

I revised again with pleasure the interesting manuscript that in many aspects has been significantly improved. The Authors were able to overcome most of criticisms.

Therefore I only have only two minor comments:

-Changes in Re-PERG were they also accompanied in parallel by changes in the so-called apparent latency or not?

-The main conclusion that Re-PERG changes may be attributable predominantly to magnocellular pathway should be softened because the other two visual stream (Konio and Parvo) were not investigated; in my opinion their study prove an alteration in RE-PERGs more evident in AD compared with VD. Otherwise it seems that the AA want to prove forcedly that an involvement of the RE-PERG means diagnosis of AD and instead may also be present in other diseases or depend from common visual changes age-related, e.g. presbyopia, cataract and so on.

7. PLOS authors have the option to publish the peer review history of their article (what does this mean?). If published, this will include your full peer review and any attached files.

**Do you want your identity to be public for this peer review?** For information about this choice, including consent withdrawal, please see our Privacy Policy.

Reviewer #1: No

Reviewer #2: Yes: Ferdinando Sartucci

We look forward to receiving your revised manuscript.

Kind regards,

Stephen D. Ginsberg, Ph.D.

Section Editor

PLOS ONE

---

## [Author Response · Author response to Decision Letter 1]

5 Jul 2020

6. Review Comments to the Author

Reviewer #1: All the concerns have been addressed. The authors corrected the manuscript in accordance with the suggestions of the reviewers, which made the manuscript suitable for publication.

Reviewer #2: Manuscript n.: PONE-D-19-31892

TITLE: RE-PERG in early-onset Alzheimer’s disease. A double-blind electrophysiological pilot study.

Article Type: Research Article.

I revised again with pleasure the interesting manuscript that in many aspects has been significantly improved. The Authors were able to overcome most of criticisms.

Therefore I only have only two minor comments:

-Changes in Re-PERG were they also accompanied in parallel by changes in the so-called apparent latency or not?

The apparent latency is evaluated by measuring the phase as a function of temporal frequency, i.e. using different temporal frequencies; in this study we used only one temporal frequency (please, see methods), therefore the apparent latency was not evaluated

-The main conclusion that Re-PERG changes may be attributable predominantly to magnocellular pathway should be softened because the other two visual stream (Konio and Parvo) were not investigated (clarified: see lines 43, 288, 299-301); in my opinion their study prove an alteration in RE-PERGs more evident in AD compared with VD. Otherwise it seems that the AA want to prove forcedly that an involvement of the RE-PERG means diagnosis of AD and instead may also be present in other diseases or depend from common visual changes age-related, e.g. presbyopia, cataract and so on. We had widely discussed problems related to the differential diagnosis between glaucoma and AD based on several diagnostic tools, including REPERG. As for the other conditions, see lines 350-355

Comments to the Author

1. If the authors have adequately addressed your comments raised in a previous round of review and you feel that this manuscript is now acceptable for publication, you may indicate that here to bypass the “Comments to the Author” section, enter your conflict of interest statement in the “Confidential to Editor” section, and submit your "Accept" recommendation.

Reviewer #1: All comments have been addressed

Reviewer #2: (No Response)

2. Is the manuscript technically sound, and do the data support the conclusions?

Reviewer #1: Yes

Reviewer #2: Yes

3. Has the statistical analysis been performed appropriately and rigorously? 

Reviewer #1: Yes

Reviewer #2: Yes

4. Have the authors made all data underlying the findings in their manuscript fully available?

Reviewer #1: Yes

Reviewer #2: Yes

5. Is the manuscript presented in an intelligible fashion and written in standard English?

Reviewer #1: Yes

Reviewer #2: Yes

6. Review Comments to the Author

Reviewer #1: All the concerns have been addressed. The authors corrected the manuscript in accordance with the suggestions of the reviewers, which made the manuscript suitable for publication.

Reviewer #2: Manuscript n.: PONE-D-19-31892

TITLE: RE-PERG in early-onset Alzheimer’s disease. A double-blind electrophysiological pilot study.

Article Type: Research Article.

I revised again with pleasure the interesting manuscript that in many aspects has been significantly improved. The Authors were able to overcome most of criticisms.

Therefore I only have only two minor comments:

-Changes in Re-PERG were they also accompanied in parallel by changes in the so-called apparent latency or not?

-The main conclusion that Re-PERG changes may be attributable predominantly to magnocellular pathway should be softened because the other two visual stream (Konio and Parvo) were not investigated; in my opinion their study prove an alteration in RE-PERGs more evident in AD compared with VD. Otherwise it seems that the AA want to prove forcedly that an involvement of the RE-PERG means diagnosis of AD and instead may also be present in other diseases or depend from common visual changes age-related, e.g. presbyopia, cataract and so on.

---

## [Editor Report · Decision Letter 2]

10 Jul 2020

RE-PERG in early-onset Alzheimer’s disease:  A double-blind, electrophysiological pilot study

PONE-D-19-31892R2

Dear Dr. Mavilio,

We’re pleased to inform you that your manuscript has been judged scientifically suitable for publication and will be formally accepted for publication once it meets all outstanding technical requirements.

Kind regards,

Stephen D. Ginsberg, Ph.D.

Section Editor

PLOS ONE

---

## [Editor Report · Acceptance letter]

14 Jul 2020

PONE-D-19-31892R2 

RE-PERG in early-onset Alzheimer’s disease: A double-blind, electrophysiological pilot study 

Dear Dr. Mavilio:

I'm pleased to inform you that your manuscript has been deemed suitable for publication in PLOS ONE. Congratulations! Your manuscript is now with our production department. 

Kind regards, 

on behalf of

Dr. Stephen D. Ginsberg 

Section Editor

PLOS ONE